# A novel optical microscope for imaging large embryos and tissue volumes with sub-cellular resolution throughout

**Gail McConnell[1]\*, Johanna Trägårdh[1], Rumelo Amor[1], John Dempster[1], Es Reid[1], William Bradshaw Amos[1,2]**

[1]Centre for Biophotonics, Strathclyde Institute for Pharmacy and Biomedical Sciences, University of Strathclyde, Glasgow, United Kingdom; [2]MRC Laboratory of Molecular Biology, Cambridge, United Kingdom

**Abstract** Current optical microscope objectives of low magnification have low numerical aperture and therefore have too little depth resolution and discrimination to perform well in confocal and nonlinear microscopy. This is a serious limitation in important areas, including the phenotypic screening of human genes in transgenic mice by study of embryos undergoing advanced organogenesis. We have built an optical lens system for 3D imaging of objects up to 6 mm wide and 3 mm thick with depth resolution of only a few microns instead of the tens of microns currently attained, allowing sub-cellular detail to be resolved throughout the volume. We present this lens, called the Mesolens, with performance data and images from biological specimens including confocal images of whole fixed and intact fluorescently-stained 12.5-day old mouse embryos.

## Introduction

During experiments with laser scanning confocal microscopes in the mid-1980s, it became obvious that the optical sectioning, which is the main advantage of the confocal method, did not work with the available low-magnification objectives, because of their low numerical aperture (N.A.) (*White et al., 1987*). In specimens such as mouse embryos at the 10–12.5 day stage, when the major organs are developing (*Kaufman, 1992*), it was impossible to see individual cells in the interior despite the lateral (XY) resolution being sufficient. Since then, stitching and tiling of large datasets has proved to be possible using computer-controlled specimen stages, but this results in a checker board pattern in the final image due to inhomogeneity of illumination and focus height errors which often cannot be corrected by software. Commercial and open source software algorithms are available to perform stitching and tiling but inhomogeneities and problems with dataset registration are clearly visible (*Legesse et al., 2015*). Because of the continuous scanning laser spot that we use with the Mesolens, we do not observe differences in fluorescence signal from one region of the image to another unless it is present in the specimen.

More recently, methods to expand the tissue volume using synthetic polymers have been demonstrated, increasing the size of the sub-micron sized structures until they can be imaged by an ordinary low-magnification, low-resolution lens, but this inevitably results in some tissue distortion (*Chen et al., 2015*). We decided that what was needed was a 4x magnification lens with an N.A. of approximately 0.5, rather than the 0.1 or 0.2 currently available, to support imaging with sub-cellular resolution through the entire specimen. Off-the-shelf lenses with the right characteristics, for example camera lenses, which could be adapted for microscopy by placing the specimen in the plane intended for the camera sensor and forming a x4 image in what would normally be the object space,

\*For correspondence:
g.mcconnell@strath.ac.uk

**eLife digest** For hundreds of years, optical microscopes have allowed living tissues to be studied in fine detail. Unfortunately, the images captured through typical microscope lenses feature a compromise between the level of detail in the image and how much of a sample can be shown. For example, densely packed individual cells often cannot be distinguished in an image that shows an entire mouse embryo.

To address this issue, McConnell et al. have developed a microscope lens called the Mesolens. This can magnify samples by up to four times in much higher detail than conventional lenses that produce the same magnification.

McConnell et al. tested the Mesolens as part of a technique called confocal microscopy, which can reconstruct the three-dimensional structure of a sample by collecting images from different layers. The resulting images allowed individual cells to be distinguished in cultures of rat brain cells. Furthermore, images of 10–12 day old mouse embryos contained enough detail to reveal many of the structures found inside cells, and allowed the distribution of cells to be tracked in developing organs such as the heart.

Ultimately, the Mesolens has the potential to be used in many different applications and could assist in the study of many biological processes. In the future, McConnell et al. will test how effectively the Mesolens works as part of other microscopy techniques.

were found to be unsuitable because of poor aberration correction including high field curvature, particularly when focused deeply into specimens immersed in fluid. The most successful attempt with a camera lens has been made recently by Zheng, Ou and Yang (*Zheng et al., 2013a*; *Ou et al., 2016*) who used a high-quality Pentax TV lens with an N.A. of approximately 0.3 to produce an image of a 10 mm diameter field with a lateral resolution of 1.56 µm. For samples such as a 10 day old mouse embryo, which is 2–3 mm thick, however, an immersion lens is needed for optically-sectioned imaging, so that the spherical aberration introduced when imaging from air and through thicker material than the thin coverslip for which the lens is designed does not destroy the optical performance as the light is focused into the specimen. Large-field of view (FOV) water-immersion lenses of great complexity have been developed for photolithography in the manufacture of semiconductor devices, but these work only with monochromatic light (e.g. at a wavelength of 193 nm) and a fixed object distance so they are not useful for biomedical imaging (*Matsuyama et al., 2006*). Fourier Ptychographic Microscopy (FPM) can give a very wide field of view image with sub-micron resolution (*Zheng et al., 2013b*). However, the depth of focus is purposefully large (~0.3 mm) in order to provide a large tolerance to microscope slide placement errors. This very large depth of focus is not suited to confocal laser scanning microscopy, where a small depth of focus is needed for optical sectioning of the imaged specimen. Also, the low N.A. lens used in wide-field FPM is for use in air only. While this is acceptable for thin tissue sections, restriction to air immersion would give dramatic fall-off in axial resolution performance at depth when imaging thick tissues because of spherical aberration, for which there are presently no correction collars or any other means of compensation. Furthermore, as noted by Zheng *et al.* the current FPM method is not a fluorescence technique. Therefore, while FPM gives high-resolution color images of thin specimens such as pathology slides, it is not suitable for fluorescence imaging of either thin or thick specimens in either 2D or 3D. Fluorescence Talbot microscopy is another method that can provide imaging over a large field of view. This usually involves placing the specimen in direct contact with the imaging sensor (*Pang et al., 2013*), which, because of the requirement for immersion and mounting of specimens in high-index material, is impractical for thick tissue imaging. The work of Pang *et al.* overcomes this limitation (*Pang et al., 2013*), and makes possible imaging of specimens spatially separated from the sensor with fluorescence contrast and an exceptionally wide field of view. However, while the lateral performance is good the reported depth of focus limit of 60 µm is too large to be useful for optically sectioning of thick tissue volumes. The imaging speed of 23 s of the instrument described by Pang *et al.* is attractive but the reported unevenness of Talbot spots over an area of 1 mm x 1 mm of 4.73% is very high when compared with both standard wide-field imaging and our wide-

field imaging using the Mesolens. Tsai et al. have performed large FOV two-photon microscopy using a commercial low-magnification objective lens (XL Fluor 4X/340, N.A.=0.28, Olympus), but they observe modest depth penetration and low axial resolution of 16 µm, as expected from the N. A. of the objective (*Tsai et al., 2015*). Furthermore, for objectives designed for multi-photon imaging, the performance (primarily colour correction) at visible wavelengths is poor or unpublished. Serial two-photon tomography can provide very high lateral resolution images of the fluorescent mouse brain (*Oh et al., 2014*). However, this imaging method requires the tissue to be destroyed because of the microtome sectioning needed and this makes repeat imaging impossible. More importantly, stitching and tiling of multiple datasets is required because of the large diameter of the tissue, and this can result in poor image registration and differences in fluorescence signal from one dataset to another. We finally note that some commercial microscopes are described as capable of imaging the same specimen from the macro-scale to the micro-scale (e.g. AZ100 Multizoom, Nikon) but at the low magnifications that permit imaging of large specimens, the N.A. of these microscopes is low, and thus the performance is as for a low N.A. objective. In this paper we present a cure for this problem. We describe a multi-immersion objective lens for wide-field epi-fluorescence and laser scanning confocal microscopy with a working distance of over 3 mm and a 6 mm FOV that is corrected for a wide range of wavelengths and demonstrate its advantages with large biological specimens.

## Results

The lens was developed from first principles with the aid of optical design software (Zemax), and optimizing for an imaging FOV of 6 mm and an N.A. as close as possible to 0.5. The FOV was chosen to match the diameter of a 12.5 day old mouse embryo specimen, and the value of N.A. was chosen based on the well-known equations (*Pawley, 2006*)

$$r_{lat} = \frac{0.61\lambda}{N.A.} \qquad (1)$$

and

$$z_{min} = \frac{2n\lambda}{(N.A)^2}. \qquad (2)$$

where $r_{lat}$ is the radius of the first dark ring around the central disk of the Airy diffraction image, $z_{min}$ is the distance of the centre of the three-dimensional diffraction pattern to the first axial minimum, $n$ is refractive index and $\lambda$ is wavelength. Since the depth of field (z resolution) is inversely proportional to the square of the numerical aperture, it rapidly becomes poor for low N.A. lenses. Below N.A. =0.45 the lateral resolution remains tolerable but the axial resolution is no longer sufficient for resolving sub-cellular details (*Figure 1—figure supplement 1*). Another important design parameter was the large working distance, to enable focusing through the embryo which, at 10 days old, is around 3 mm thick. This dictated that the lens elements should be physically large, to accommodate the wide collection angles for light originating at a large distance from the lens.

Because embryos may be examined in a variety of optically dissimilar fluids such as water, glycerol, oil and benzyl benzoate (such as for the clearing liquid BABB), it was necessary to make the lens suitable for use with different immersion fluids and mounting media. The Mesolens was designed for immersion into non-corrosive immersion media such as oil (Type DF), water and glycerol. We use BABB only as a mounting medium, and the specimen and BABB are separated from the immersion fluid by a type 1.5 coverslip. This meant providing means to shift lens element groups during use so that adequate correction for spherical and other aberrations could be obtained for the different liquids. In practice, the necessary shift was controlled by two knurled rings on the lens barrel, similarly to a standard microscope objective that corrects for the coverslip thickness or immersion fluid. In addition, a further interchangeable flat compensator plate was incorporated into the design to compensate for spherical aberrations introduced by the different immersion fluids, with a thickness specific to the immersion medium used. The main challenge in our lens design was to achieve the colour correction between 400 nm and 700 nm (see *Figure 1—figure supplement 2*). Also, the diverse glass types required all had to be tested for auto-fluorescence, and the precise

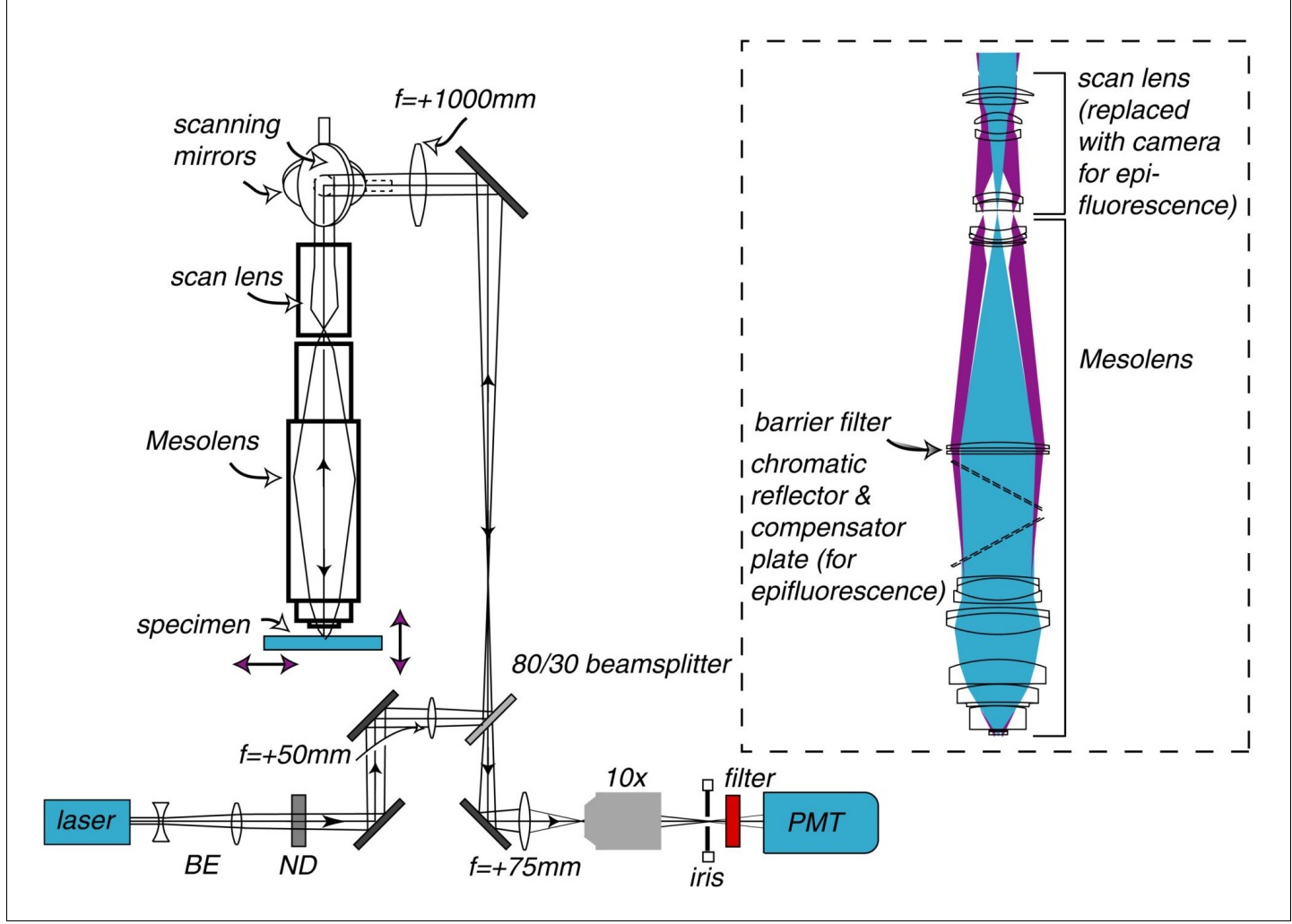

**Figure 1.** Schematic diagram of the laser scanning confocal Mesolens system. BE=beam expander, ND= neutral density filter. Only one detection channel is shown here for ease of presentation. Inset: Illustration of the optical train of the Mesolens and scan lens. Also shown is the position of the compensator plates and camera when the Mesolens is used for wide-field epi-fluorescence imaging.

The following figure supplements are available for figure 1:

**Figure supplement 1.** Lateral and axial resolution of a dry lens as a function of N.A.

**Figure supplement 2.** Illustration of the excellent chromatic correction of the Mesolens.

refractive index of each selected sample had to be ascertained and the lens design adjusted to accommodate the small variations from one melt to another of nominally the same glass.

The Mesolens is shown in schematic form in *Figure 1*. In total the Mesolens comprises of 15 optical elements of up to 63 mm in diameter when used in laser scanning confocal mode. This increases to 17 elements for wide-field epi-fluorescence mode, where a (removable) chromatic reflector with a 30 degree angle of incidence with a custom mercury-line coating designed specifically for this angle of incidence (Chroma Technology Corp.) was incorporated into the optical train to introduce the incoherent light source, as shown in the insert to *Figure 1*. A tilted compensating plate, similar to the chromatic reflector but anti-reflection coated only, was added to eliminate astigmatism introduced by the chromatic reflector (Chroma Technology Corp.). The lens design required manufacture and mounting of the optical elements to fine tolerances, e.g. centration to better than 3 µm. This was performed by IC Optical Systems Ltd.

For wide-field camera imaging in brightfield and epi-fluorescence modes, a camera with a 35 mm sensor chip was placed directly on top of the Mesolens via an F-mount. The excitation and detection beam path for laser scanning confocal microscopy is illustrated in *Figure 1* and described in the Materials and methods section.

With elements of up to 63 mm in diameter and a scan lens with an aperture diameter of 30 mm the Mesolens cannot be fitted into an ordinary microscope frame and when used in scanned image modes such as confocal, the scanning mirrors have to be very large. We used custom mirrors together with a galvo scanner (8360K, Cambridge Technology) which were made from lightweight beryllium and back-thinned to reduce the mass. These mirrors were required to be flat to λ/10 for sufficiently small aberrations. A multi-element scan lens made to our design by Beck Optronic Solutions was used to image the back focal plane of the Mesolens into an aperture of 30 mm diameter. The scan lens consisted of 6 optical elements, and was corrected for astigmatism, spherical aberration and chromatic aberrations for wavelengths between 400 nm and 750 nm.

We sawed through an upright microscope (Optiphot, Nikon) to remove all but the stage, substage condenser, illuminator and base. We positioned this underneath the Mesolens and added a computer-controlled z-positioning system (Optiscan II, Prior Scientific). The condenser lens was of smaller aperture than the Mesolens and although it had a high-enough N.A. it could not fill the field of the Mesolens: for this, a new type of condenser would have to be designed. Nevertheless, it allowed for some basic optical transmission imaging.

We note that with a 6 mm FOV the number of pixels required for a full Nyquist sampling of the image with the design lateral resolution of 0.6 μm is 20000 x 20000 pixels. Using a minimum

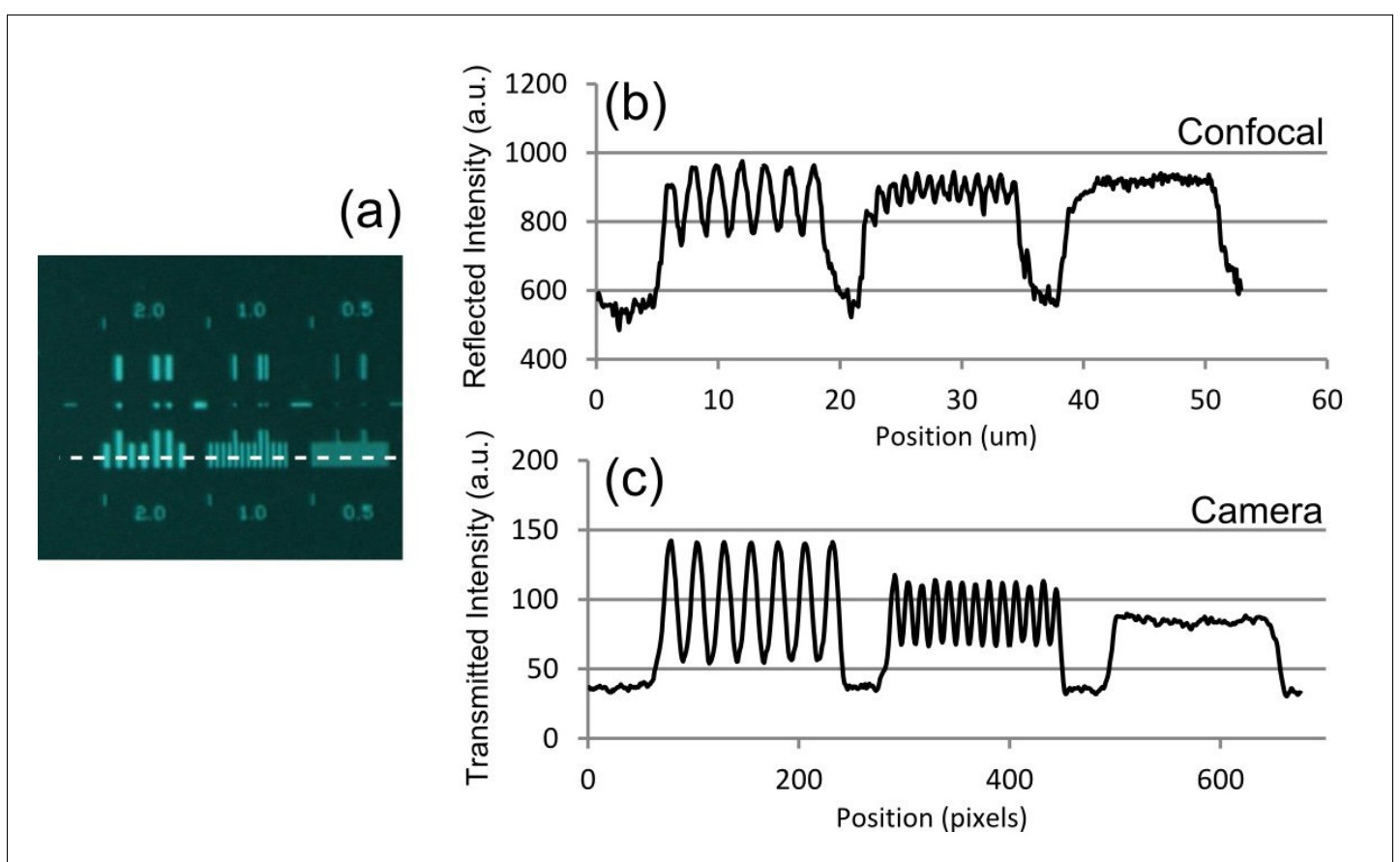

**Figure 2.** Optical resolution measurements for the Mesolens. (a) shows an image of the specimen consisting of a series of gratings with periods of 2 μm, 1 μm, and 0.5 μm obtained with brightfield illumination and captured in wide-field mode using a camera. (b) and (c) show intensity linescan measurements of this region of the specimen obtained in (b) laser scanning confocal reflection mode using a 514 nm laser and (c) in camera mode. The intensity profiles in (b) and (c) are obtained from the region indicated by the dashed line in (a).

practical pixel dwell time of ca. 0.5 μs this leads to an acquisition time of 200 s for a full FOV, full resolution image. This is similar to the time needed to image 6 mm x 6 mm using multiple images from a 10x objective with N.A.=0.4 yielding a similar z-resolution, and stitching and tiling them. With our imaging system, however, we circumvent the problems of image alignment. In this case the mirror speed is a mere 92 Hz, which is at the limit of what such large mirrors can achieve. In summary, despite appearance, this imaging system is not slower in data rate than a standard laser scanning confocal microscope. For imaging in wide-field camera mode, full Nyquist sampling would require a camera with 400 megapixels.

To summarize, the lens specifications were a magnification of intermediate image of 4x, an N.A. of 0.47, with a working distance of 3.15 mm, 3.40 mm, and 3.74 mm for water, glycerol, and oil immersion media respectively. The distortion was designed to be <0.7% at the periphery of the field. The lens was colour-corrected over the full visible range to allow imaging of samples with multiple fluorescent labels. The flatness of field was designed to be <3 μm for wavelengths of 400–700 nm across the 6 mm FOV.

## Measurements of optical performance

For quantitative measurements of resolution, a special test target was prepared by electron beam lithography of a 75 nm thick nickel-chromium coating on glass (*Oldenbourg et al., 1993*). The lateral image resolution was judged by the visibility of a series of gratings with periods of 2 μm, 1 μm, and 0.5 μm. In brightfield transmission mode using wide-field camera detection and in laser scanning confocal reflection mode using the 514 nm laser with photomultiplier tube (PMT) detection, the resolution was found to be better than 1 μm (*Figure 2*), with the 0.5 μm grating unresolved but the 1 μm grating clearly resolved.

To evaluate the full point-spread function (PSF) we used yellow/green-emitting fluorescent beads. For comparison similar measurements were conducted with a conventional dry objective lens of similar magnification (5x/0.15 N.A HCX PL Fluotar, Leica Microsystems) mounted on an upright microscope. The results are presented in *Table 1*. The bead diameter ($d_{bead}$) was deconvolved using

$$d_{measured} = \sqrt{d_{bead}^2 + d_{real}^2}. \tag{3}$$

The lower than theoretical diffraction-limited resolution in z for laser scanning confocal mode was likely due to residual (primarily astigmatic) aberrations from the scan lens or from inaccuracies in the scan-lens-to-Mesolens separation. The astigmatic error was visible at the edges of the FOV. Nevertheless, the z resolution is, as we will demonstrate below, sufficient to resolve sub-cellular detail, and far superior to a standard objective lens with comparable magnification.

To investigate whether the lower than theoretical diffraction-limited axial resolution was associated with the scan system and Mesolens, or arose only from the Mesolens, we performed measurements of the PSF in wide-field epi-fluorescence mode. The results are presented in *Table 1* and shows that the z-resolution is 7 μm, which is only somewhat higher than the theoretical resolution (6.3 μm) and there are some residual astigmatic aberrations. The lack of improvement of the axial resolution in laser scanning confocal mode is therefore likely due to aberrations in the scan lens and illumination system. Nevertheless, the laser scanning confocal mode still achieves optical sectioning

**Table 1.** Measured and calculated values (μm) for the FWHM of the PSF in laser scanning confocal mode and wide-field epi-fluorescence mode for a conventional 5x/0.15 N.A. lens and the 4x/0.47 N.A. Mesolens used with oil immersion. We have assumed emission λ=550 nm for the calculations.

| | 5x/0.15 N.A. | | Mesolens (4x/0.47 N.A.) | |
|---|---|---|---|---|
| | Theoretical | Measured | Theoretical | Measured |
| XY FWHM, epi | 1.9 | 1.9 | 0.6 | 0.7 |
| Z FWHM, epi | 41 | 51 | 6.3 | 7 |
| XY FWHM, confocal | 1.8 | 1.7 | 0.6 | 0.8 |
| Z FWHM, confocal | 25 | 38 | 3.7 | 8 |

not present with wide-field epi-fluorescence mode. The flatness of field was measured to be <=3 µm over the central 4.2 mm of the FOV in reasonable agreement with the design specifications.

In order to make measurements of the relative optical throughput efficiency of the Mesolens as compared with a commercial low-magnification, low numerical aperture lens (4x/0.1 N.A. Plan, Nikon) a light-emitting diode (HLMP-2855, Broadcom) was used as a specimen of standard luminosity (*Beach and Duling, 1993*). The LED was driven at constant current and the intensity was monitored using a photodiode. The specimen was set up with the same CCD camera for the Mesolens in oil-immersion mode and the 4x/0.1 N.A. objective. Using the same exposure time the intensity with the Mesolens was found to be 25x that with the conventional objective, in good agreement with the 4.7 times higher N.A. available from the Mesolens. To measure the optical throughput we compared the laser power immediately after the beam splitter in the periscope (see *Figure 1*) with the laser power at the specimen plane at a wavelength of 488 nm, and we measured an on-axis transmission of 11%. A significant loss results from overfilling the top mirror in the periscope, which is required in order to achieve the flattest possible wavefront. There are also additional losses from the scanning mirrors and in the scan lens. Although the optical throughput of a standard microscope objective can be several tens of percent, the transmission of the complete system is more similar to what we measure here. We also note that the transmission value of the complete system is rarely given explicitly by microscope manufacturers but, based on the quoted power of the laser system (e.g. 50 mW for a Krypton-Argon laser operating at a wavelength of 488 nm) and measuring the maximum average power at the specimen plane (e.g. 3 mW after the objective lens), we measure a similar magnitude of optical loss with commercial confocal microscopes compared to the Mesolens. The good transmission of the Mesolens itself is also confirmed by our comparison of the transmission of the Mesolens with that of a 4x/0.1 N.A. lens in camera mode.

## Imaging of biological specimens

*Figure 3* and the *Video 1* show wide-field epi-fluorescence data from an explant culture of rat embryonic brain as a colour merge of three camera images at the same focus position. In this

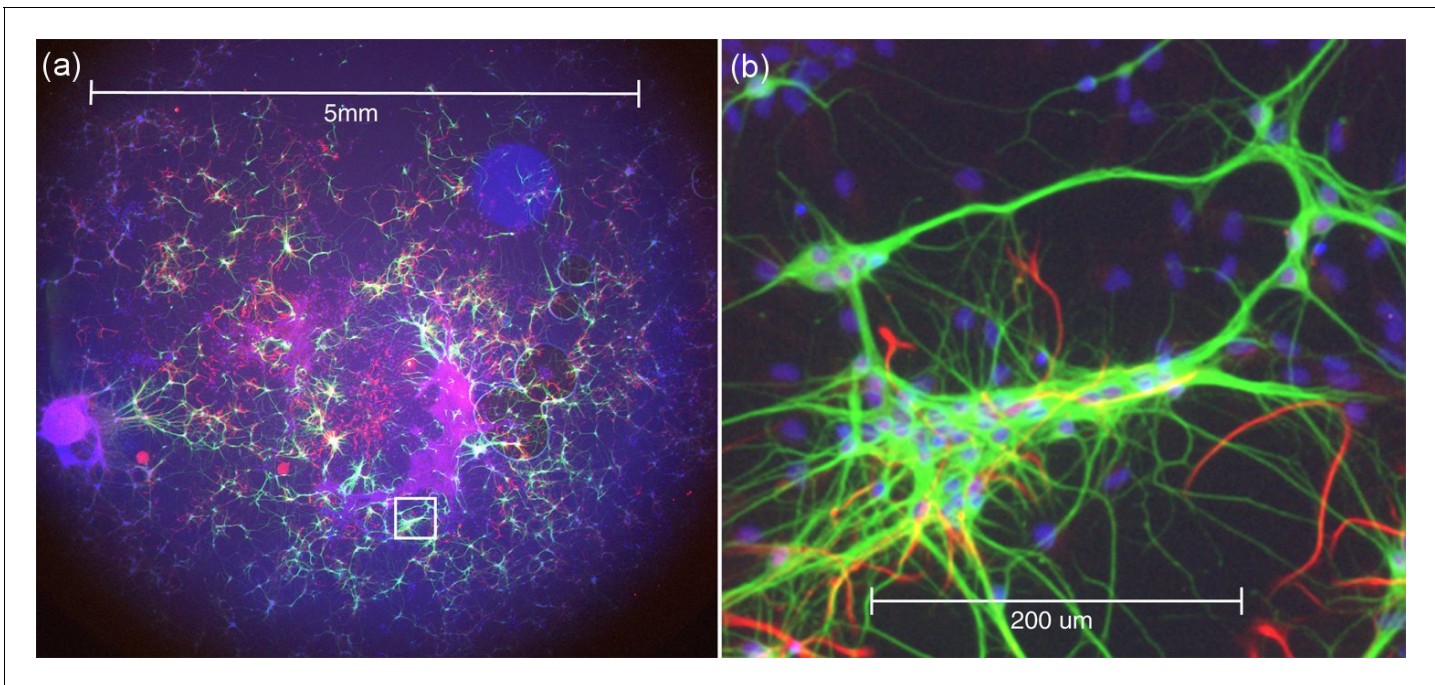

**Figure 3.** Wide-field epi-fluorescence imaging of an explant culture of rat embryonic brain as a colour merge of three camera images at the same focus position, demonstrating the excellent colour correction of the Mesolens. The blue fluorescence is of nuclei stained with DAPI, neurons show green because of Alexa 488 conjugated to an antibody against beta-III tubulin and astrocytes are red because of Alexa 546 conjugated to anti-GFAP. (a) Full FOV (b) Software zoom of the area indicated in (a) (*Video 1*) to visualize individual nuclei and neurons.

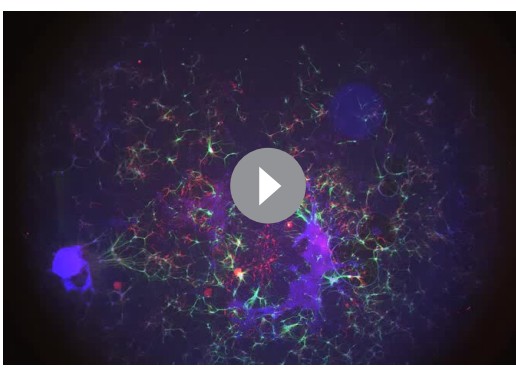

**Video 1.** Epi-fluorescence in an explant culture of rat embryonic brain as a colour merge of three camera images at the same focus position. The blue fluorescence is from nuclei stained with DAPI, neurones show green because of Alexa 488 conjugated to an antibody against beta-III tubulin and astrocytes are red because of Alexa 546 conjugated to anti-GFAP. The zoom is generated by software only. Halfway into the movie we have changed to a smaller format camera mounted on an image magnifier to achieve fine enough sampling. Note that this will not enhance the optical resolution of the image but was necessary to overcome the lack of a camera able to record hundreds of megapixels in a single image.

preparation, the blue fluorescence is of nuclei stained for DNA with DAPI, neurones show green because of Alexa 488 conjugated to an antibody against beta-III tubulin and astrocytes are red because of Alexa 546 conjugated to anti-GFAP (glial fibrillary acid protein). A custom multiband reflector (Chroma) was used with a multiband barrier filter and, for each image, a different excitation waveband was selected from a mercury arc lamp. This image clearly shows the excellent chromatic correction of the lens with all three colours simultaneously in focus. Furthermore, the image demonstrates the possibility to view both the full large FOV and tiny details in the same image, using only (17x) software zoom.

To demonstrate the superior sectioning capability of the Mesolens compared to a standard objective lens with comparable magnification, we imaged the same fixed, cleared and acridine orange-stained 10 day old mouse embryo with the Mesolens and with a dry objective lens of similar magnification (5x/0.15 N.A. HCX PL Fluotar, Leica Microsystems) mounted on an upright laser scanning microscope. The embryo was lying on its side so that XY scanning produced an optical section that was anatomically sagittal. The XZ images are displayed in *Figure 4a and b*. *Figure 4a* was taken with a z-plane spacing of 1.5 μm and a line average of n=12 and *Figure 4b* has a plane spacing of 1 μm and no averaging. For both lenses, but acutely for the 5x/0.15 N.A. lens, this is much smaller than the size of the PSF. The Mesolens can image throughout the entire 2 mm thick embryo with sub-cellular resolution (we can see individual nuclei all the way through). This is in contrast to the image with the commercial low magnification, low N.A. objective, where the long needle-like vertical structures indicate a very poor z resolution. The lateral resolution also remains micron-sized throughout the embryo, as can been seen from *Figure 4c*, which presents an XY cross section midway through the same embryo. In *Figure 4b* the dark cavity within the midbrain on the left and the neural canal within the spinal cord on the right are imaged with good axial resolution even though they are deep within the embryo, which is equally-well resolved at the top and bottom of the z range. Although the larger cavities are also visible in the image taken with the 5x lens, finer details such as the dark extracellular spaces in the bright neural tube to the right of the image and the epithelial cell monolayer (upper right margin and left lower corner) are only visible in the Mesolens image (*Figure 4b*). In the XY section (*Figure 4c*) similar fine details including individual cell nuclei in all the tissues are clearly resolved. Structures within the embryo were identified by means of (Kaufman, 1992). The constant signal with increasing depth throughout the specimen arises from having a well-cleared specimen. We did not adjust the laser power or detector gain when imaging deeper into tissue.

To demonstrate the capability of the instrument as a two-channel detection system, and to illustrate the very large FOV, we imaged 12.5 day old embryos, which cover nearly the full 6 mm FOV. This embryo was stained with an antibody against axonal beta tubulin coupled to the fluorescent dye Alexa 594, to highlight nerve cells and with acridine orange, here used chiefly as a nuclear marker. A dual-colour single optical section of a 12.5 day embryo stained following this protocol is presented as *Figure 5*, with a 3D rendering around one of the eyes of the embryo shown as *Video 2*. A lower-quality, smaller version of the same data is available as *Video 3*. In the 3D rendering there are 60 images axially separated by 3.33 μm, imaging a total thickness of 200 μm and we used the fluorescence mode of our image reconstruction software (Volocity, Perkin Elmer) which applies a direct opacity rendering. We can clearly identify fine structures throughout the embryo such as the

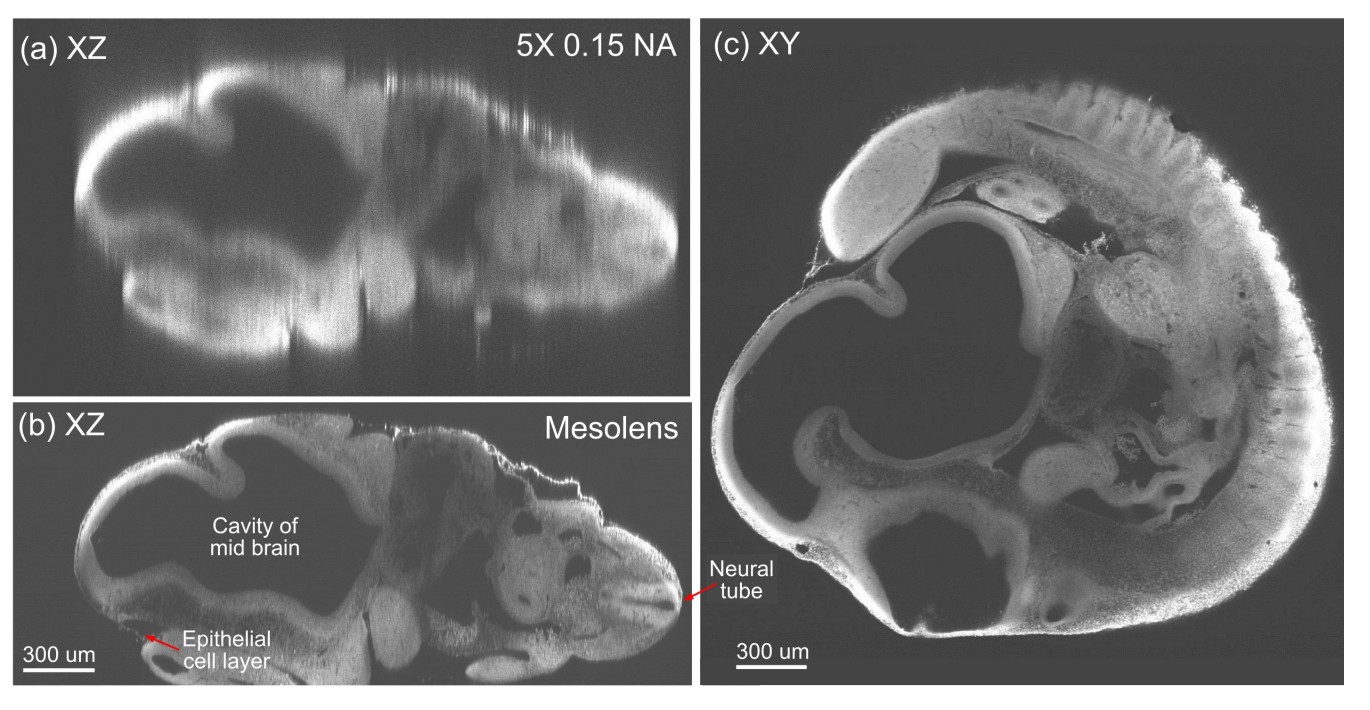

**Figure 4.** Optical sectioning of an optically-cleared and fluorescently-stained 10 day old mouse embryo. (a) is the embryo imaged in XZ using a dry objective lens of similarly low magnification to the Mesolens (5x/0.15 N.A., HCX PL Fluotar, Leica Microsystems). (b) is the same embryo imaged at the same region in XZ using the Mesolens with oil immersion. The Mesolens can image throughout the entire 2 mm thick embryo with sub-cellular resolution: individual nuclei can be observed all the way through. This is in contrast to the image with the commercial low magnification, low N.A. objective, where the long needle-like vertical structures indicate a very poor z resolution. *Figure 4c* shows an XY cross section at a depth of around 1 mm into the same specimen imaged using the Mesolens. Full resolution versions of (b) and (c) are available as *Figure 4—figure supplements 1* and *2*.

The following figure supplements are available for figure 4:

**Figure supplement 1.** Optical sectioning of an optically-cleared and fluorescently-stained 10 day old mouse embryo imaged using the Mesolens.

**Figure supplement 2.** Optical sectioning of an optically-cleared and fluorescently-stained 10 day old mouse embryo imaged using the Mesolens.

developing heart muscle fibers and fine details in the eye such as the corneal endothelium. The inset shows a blow-up of the eye region revealing the individual cell nuclei. We emphasize that this image is only a software zoom from the full scan range of the confocal Mesolens system, not a smaller scanned region. In the eye, the lens was visible as a slightly absorbing structure with a sharply defined outline, surrounded in turn by cells within the aqueous humour and then the strongly-stained retina. All these structures were enclosed in the corneal stroma, lined on the inside with the corneal endothelium and on the outside with the epithelial precursor of the conjunctiva.

To demonstrate that an outstanding axial resolution is retained over the full FOV with only a small degradation at the edges of the image we performed a two-color XZ image of the 12.5 day old embryo. This is presented in *Figure 5—figure supplement 1*.

## Discussion

We have shown here that it is possible, using known design software and fabrication to exact tolerances, to design and construct an objective to facilitate imaging of objects up to 6 mm wide and 3 mm thick in their entirety with depth resolution of only a few microns instead of the tens of microns currently attained, allowing sub-cellular detail to be resolved. We call this lens the Mesolens. In doing this work we have departed from a design principle of more than a century, which is that the number of lateral resolution units in the field diameter should be approximately the same for all

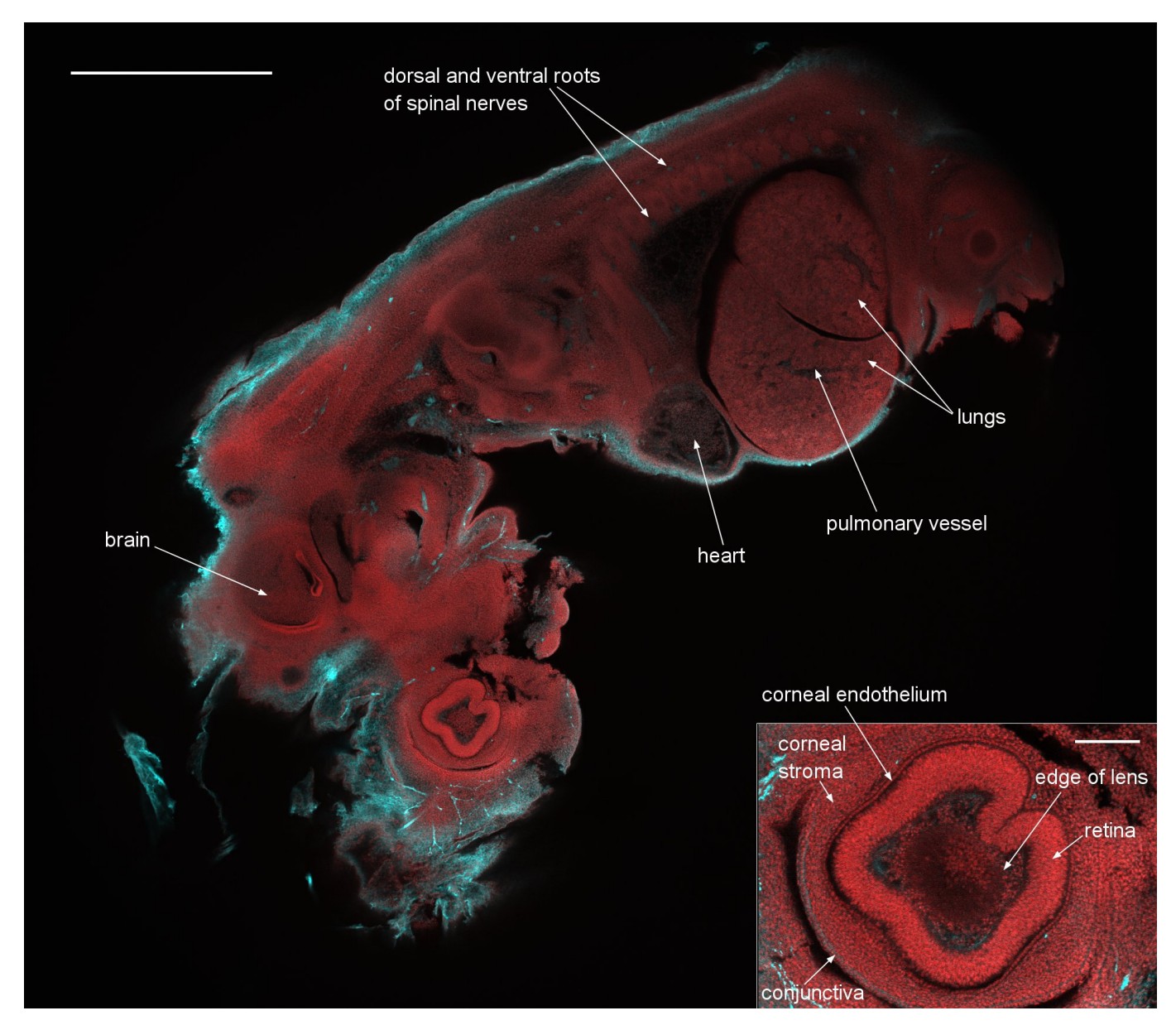

**Figure 5.** A dual-colour optical section of a 12.5 day old mouse embryo obtained with the Mesolens in laser scanning confocal mode. The neuronal axons are stained with Alexa 594 (cyan) and nuclei with acridine orange (red). Scale bar=1 mm. The inset shows a blow-up of the eye region revealing the individual cell nuclei. The image plane of this inset is located 36 µm closer to the specimen surface than in the full image shown. The scale bar on the inset image is 300 µm. We can clearly identify fine structures throughout the embryo such as the developing heart muscle fibers and fine details in the eye such as the corneal endothelium. The cyan streak in the lower right corner of the blow-up indicates a stained nerve. A z-stack of the same specimen is presented in *Video 2*.

The following figure supplement is available for figure 5:

**Figure supplement 1.** Illustration of the high z-resolution throughout the full 6 mm FOV.

objectives. As we have shown here, the lateral resolution in the image field of this new lens is too great for the human eye, but suits cameras of the highest pixel number available or laser scanning confocal operation and, crucially, the axial resolution justifies the collection of several hundred optical sections of a thick specimen within the working depth. We have shown that single cells, including

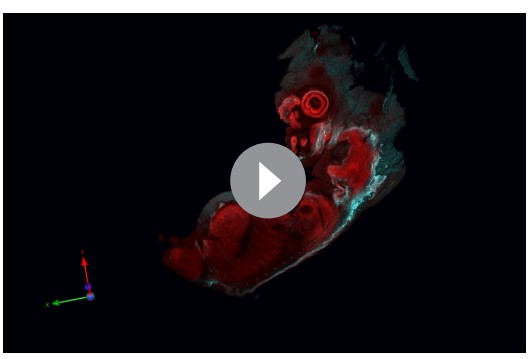

**Video 2.** The movie is a 3D rendering generated from a z-stack through the eye of a 12.5 day old mouse embryo (the same embryo as in *Figure 5*). This embryo was stained with an antibody against axonal beta tubulin coupled to the fluorescent dye Alexa 594, to highlight nerve cells, and with acridine orange, here used chiefly as a nuclear marker. All frames in the rendering are from full 6 mm x 6 mm frames and only software zoom is used. The blow-up of the eye region reveals individual cell nuclei. The lens was visible as a slightly absorbing structure with a sharply defined outline, surrounded in turn by cells within the aqueous humour and then the strongly-stained retina. All these structures were enclosed in the corneal stroma, lined on the inside with the corneal endothelium and on the outside with the epithelial precursor of the conjunctiva.

the nucleated erythrocytes of mouse embryos, and syncytial heart muscle fibres can be seen not just near the surface as with conventional objectives but throughout the depth of the embryo. A biologist can therefore discover whether, for example, a defect in development involves changes in the shape of a deep-located organ, or of cell shape or cell number, or of the nucleus-to-cytoplasm ratio. No existing microscope can show all of these features simultaneously in an intact mouse embryo in a single image.

With the Mesolens we also have the possibility of switching from a relatively slow laser scanning mode to fast camera mode with the large field of view in the same focal plane. This is not possible with higher magnification objective lenses. This is of value both in searching for rare image features, and in adjusting the compensation collars of the objective lens.

We do not observe difficulty with the penetration of the acridine orange stain for the E10 embryos, although for the thicker 12.5 day old embryos this might be part of the reason for the reduced signal in the centre of the embryo seen in *Figure 5—figure supplement 1a*.

If the volume of the point spread function of an objective lens to the first ellipsoidal minimum is divided into the total accessible 3D volume a measure of the number of resolved voxels can be obtained (a quantity comparable to the Nyquist sample but for a 3D volume). The maximum size of a Mesolens dataset is approximately 200x that for a 100x/1.4 N.A. lens. Calculated from the data presented here for the Mesolens, a complete image sampled at Nyquist resolution over the 6 mm x 6 mm x 3 mm volume generates a 639 GB digital image, which is large but the capacity and speed of current computers can readily handle data files of this size.

The ability to record such high-volume data makes it likely that many applications other than the imaging of mouse embryos will be found. Provided the problem of tissue transparency can be solved, there are obvious applications in neurobiology. Also, the Mesolens can reveal spatially rare events, and may have application in the measurement of the mitotic index in slowly-reproducing tissues. The 25x higher optical throughput could also facilitate imaging of weakly emitting specimens, for example specimens expressing fluorescent proteins at a low level or containing single cells bearing luciferase as a bioluminescent marker.

Recently, a lens of similar numerical aperture to the Mesolens (N.A.=0.6) with a field of 5 mm diameter and a collection depth of 1 mm has been described (*Sofroniew et al., 2016*). However, this lens is corrected only for a narrow wavelength band (900–1070 nm) and is restricted in use to single-spot multi-photon imaging, whereas the Mesolens can serve as a multi-wavelength camera lens, a confocal objective, and

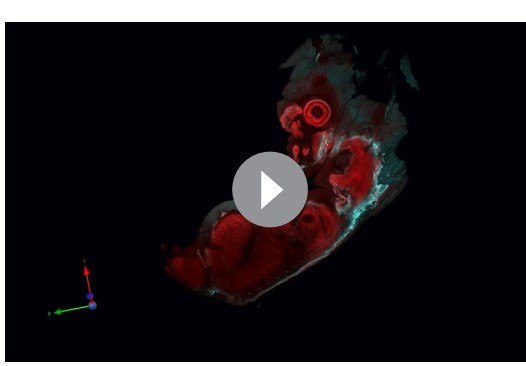

**Video 3.** This movie is a highly compressed and lower quality version of *Video 2*. *Video 3* has a much smaller file size for quick downloading and viewing.

potentially also as a multi-photon objective, both single-spot and wide-field (*Amor et al., 2016*).

This development represents the most radical change in microscope objective design for over a century, and we believe that despite the fact that the Mesolens cannot be fitted to a conventional microscope stand, it has the potential to transform optical microscopy through the acquisition of sub-cellular resolution 3D datasets from large tissue specimens.

## Materials and methods

### The Mesolens

The excitation and detection beam path for laser scanning confocal microscopy with the Mesolens is illustrated in *Figure 1*. The excitation light was provided by a fiber-coupled multi-line laser system (Laserbank, Cairn Research), comprising 405 nm, 488 nm, 514 nm, 532 nm, 561 nm and 637 nm wavelengths. The laser was expanded using an f=−25 mm lens (LC1054, Thorlabs) and an f= +125 mm lens (LA1986-A, Thorlabs), propagated through an f=+50 mm lens (LA1131-A) and directed towards the scanning system using a periscope. An 80/30 beamsplitter (PBSW-532, Thorlabs) formed the lower mirror of the 1 m periscope, and the top mirror (Thorlabs BB2-E02) steered the beam towards a large-aperture f=+1000 mm achromatic doublet lens (LA1779-A-ML, Thorlabs) to collimate the light before reaching the scan mirrors. The scanning mirrors directed the light through the scan lens and the Mesolens focused the light into the specimen.

The reflected light or fluorescence signal from the specimen was propagated back through the Mesolens, scan lens, scanning system and periscope, and was transmitted through the beamsplitter, chosen to transmit 80% of the (unpolarised) fluorescence emission and reflect 30% of the linearly polarized laser beam. The signal was collected using an f=+75 mm lens (LA1608-A, Thorlabs) and focused through the imaging iris using a non-infinity corrected 10x objective lens (10 OA 25, Comar Optics). The total magnification of the system was 1440, and the confocal iris was set to 1 Airy unit = 1.7 mm. A two-channel PMT system was built for two-colour imaging, with each PMT having its own imaging iris. The PMT used for the images in *Figures 2b* and *4b–c*, and *Figure 4—figure supplements 1*, *2* and *Figure 5—figure supplement 1* was a PMM02, (Thorlabs) and for *Figure 5* and *Video 2* a Senstech P30-09 was used for the red channel and Senstech P30-01 was used for the green channel. The bandpass filters were chosen to match the emission wavelengths of the fluorescent dyes in the specimen (and were removed for reflection imaging).

Images were digitized using a PCI-6110 (National Instruments) multifunction DAQ card with a 12 bit, 5 MHz, simultaneous sampling A/D converter and 16 bit, 4 MHz, D/A converter. For optimum scanning rate and stability, a sine wave scan waveform (max. 66 Hz), was used, with the first half of each cycle digitized and remapped to obtain a linear image field. The system was controlled using an in-house, laser scanning software package, 'Mesoscan', designed to handle 400 megapixel scanned images. Images were stored in the Open Microscopy Environment OME.TIFF format and analysed using ImageJ.

### Resolution measurements

The camera used to capture the bright field image in *Figure 2* was a Nikon Ds-Qi1Mc mounted on top of an image magnifier with a 10x objective lens serving as the magnifying lens. Note that this will not enhance the optical resolution of the image but is a way around the lack of a camera with sub-micron pixel size as required for Nyquist sampling of the image. The same method is used for measuring the PSF using fluorescent beads in epi-fluorescence mode. The upright microscope used for comparing the resolution of the Mesolens to that of a low N.A. 5X lens was a Leica DM6000, with an SP5 scanning unit (Leica Microsystems).

Wide-field epi-fluorescence measurements were made by using 470 nm from a mercury short–arc source for excitation. The point-spread function of the Mesolens in laser scanning confocal fluorescence mode with oil immersion was measured using 488 nm excitation. The yellow/green-emitting beads used for the resolution measurements were 500 nm diameter beads (17152–10, Polysciences Inc.) for the lateral resolution measurements in laser scanning confocal mode, and for the lateral and axial resolution measurements in wide-field epi-fluorescence mode for the Mesolens, and 1 μm diameter beads (18860–1, Polysciences Inc.) for the remaining resolution measurements respectively. To immobilize the beads they were dried down on to cleaned coverslips and immersed in a Gelvatol

medium (341584, Sigma-Aldrich) containing DABCO (diazo-bis-cyclo-octane, 290734, Sigma-Aldrich) as antifade. For the measurement in wide-field epi-fluorescence mode we used a 10x image magnifier as described above and a cooled CCD camera with 6.45 μm pixel size (CoolSnap HQ2, Photometrix).

The flatness of field was measured in epi-fluorescence in oil immersion mode using 6 μm fluorescent beads (18862, Polysciences Inc.) dried down on the surface of a mirror serving as a flat substrate. A camera with a 35 mm sensor chip (C3900-024, Hamamatsu) was placed directly on top of the Mesolens via an F-mount, and images were captured using the manufacturer's software (HCImage, Hamamatsu).

## Explant culture of rat brain, *Figure 3*, *Video 1*

Explant cultures of embryonic rat brain were prepared and fixed with 2% paraformaldehyde and stained as follows: The nuclei were stained with DAPI, the neurons were stained with Alexa 488 conjugated to an antibody against beta-III tubulin and astrocytes were stained with Alexa 546 conjugated to anti-GFAP. For the fluorescence imaging with this specimen a mercury lamp was used for excitation, with custom filters designed for DAPI, Alexa 488 and Alexa 546 (Chroma). Two different cameras were used to image the same region (*Figure 3*, *Video 1*). A low-power image was made by recording the entire field of the Mesolens on a monochrome 35 mm CCD chip with 10 megapixels, each of 9 μm square (C3900-024, Hamamatsu). Three images, using UV, blue or green excitation from the mercury lamp were merged and coded blue, green and red respectively. A high-power image was made with a small format color camera Nikon DSL1 (used without a magnifier, in contrast to the resolution measurements described above), recording only a 2.2 mm x 1.6 mm portion of the Mesolens field with a square pixel size of 3.4 μm. The theoretical resolution of the Mesolens (0.6 μm) corresponds to 2.4 μm in image space in the camera, so neither camera allowed Nyquist sampling. In the *Video 1*, the wide-field epi-fluorescence image is first software-zoomed and then replaced by the image from the smaller format camera, which is further software zoomed and the view is software-panned around the region recorded in the small-format camera. This procedure was necessary to overcome the lack of a camera able to record hundreds of megapixels in a single image.

## Protocols for staining embryos (*Figures 4* and *5*, *Figure 4—figure supplements 1*, *2* and *Figure 5—figure supplement 1*, *Video 2* and *Video 3*)

For the 10 day old embryos in *Figures 4*, *Figure 4—figure supplements 1* and *2*, embryos were dissected from mouse uteri, fixed in ethanol: acetic acid 3:1 by volume overnight at 4°C and then transferred into aqueous media via a series of ethanol/water mixtures with decreasing ethanol content. The embryos were then stained overnight with acridine orange (0.05% in phosphate-buffered saline), rinsed overnight in PBS, dehydrated with ethanol solutions, cleared in xylene and mounted in Histomount (Fisher Scientific).

For the embryo presented in *Figure 5* and *Figure 5—figure supplement 1* embryos were dissected from mouse uteri, fixed in Dent's Fix for 24 hr at 4°C, washed 3 times in PBS and then bleached in Dent's Bleach (for 24 hr) at 4°C. Embryos were then rinsed 5 times in methanol, and fixed in Dent's Fix (for 24 hr) at 4°C and stored at 4°C until staining. For antibody staining, embryos were rinsed 3 times in PBS, then washed 5 times in PBS for 1 hr per wash at room temperature. Mouse anti-Tubb3 (beta-III-tubulin) primary antibodies (Sigma-Aldrich T5076-200UL; 1:250 concentration) were applied in blocking solution (Bovine Serum Albumin (BSA)), overnight at room temperature, after which the embryos were rinsed 3 times in PBS, then washed 5 times in PBS, for 1 hr per wash at room temperature. Subsequently, rabbit anti-mouse secondary antibodies (1:250) conjugated to Alexa 594 (Invitrogen) were applied in blocking solution (BSA), overnight at room temperature, after which the embryos were rinsed 3 times in PBS, then washed 5 times in PBS, for 1 hr per wash at room temperature.

For optical clearing, embryos were first dehydrated by removing half of the PBS and replacing with methanol for 5 min. They were then washed 3 times in methanol for 20 min per wash and then half of the methanol was removed and replaced with 1:2 *benzyl alcohol:benzyl benzoate* (BABB) for 5 min. Finally, embryos were cleared in 100% BABB overnight at 4°C and were stored in BABB at room temperature until re-staining with acridine orange. All the washing, staining and clearing steps

were done with the sample on a rocker. The embryos were then stained with nuclear stain acridine Orange as follows. The embryos were transferred from BABB to PBS via a series of solvents. First, the embryo was transferred to xylene, and left to soak in several changes of xylene to remove the BABB over a period of days. It was then transferred from xylene to three changes of absolute ethanol, then 90% ethanol, 10% water by volume, then 75% ethanol, 50% ethanol and finally into distilled water (with at least 10 min in each change). Acridine orange powder was dissolved in PBS to a pale yellow solution. The embryo was left in the staining solution overnight with gentle rolling on a slow turntable. The embryo was then washed in two changes of PBS over an hour and passed through the same series of alcohols up to xylene and then rolled in BABB overnight. These embryos were imaged in BABB.

### Imaging parameters for the embryos (*Figures 4*, *5*, *Figure 4—figure supplements 1*, *2* and *Video 2*)

The images in *Figures 4b,c*, *Figure 4—figure supplements 1* and *2* were acquired using 488 nm excitation with a power of ca 100 µW at the sample plane and the signal was detected using a 525/39 nm band pass filter. For *Figure 4b*/*Figure 4—figure supplement 1* we used a pixel size of 1 µm x 1 µm (XZ) a scan speed of 22 lines per second and no frame averaging. For *Figure 4c*/*Figure 4—figure supplement 2* we used a pixel size of 0.5 µm x 0.5 µm (XY) a scan speed of 22 lines per second and a frame average of 4. The low line speed was necessitated by the low bandwidth of the PMT used for the images.

The z-stack in *Figure 5*/*Video 2*/*Video 3* was acquired using 488 nm excitation with a power of ca 50 µW at the sample plane and the signal was detected using a 525/39 nm band pass filter for imaging the acridine orange. We used 561 nm excitation with a power of approximately 150 µW at the sample plane and the signal was detected using a 600 nm long pass filter for imaging Alexa 594. We used a pixel size of 1 µm x 1 µm x 3.33 µm (XYZ), a scan speed of 45 lines per second and a frame average of 4.

The image for *Figure 5—figure supplement 1* was acquired using 488 nm excitation with a power of ca 50 µW at the sample plane and the signal was detected using a 525/39 nm band pass filter for imaging the acridine orange. We used 561 nm excitation with a power of approximately 150 µW at the sample plane and the signal was detected using a 600 nm long pass filter for imaging Alexa 594. We used a pixel size of 1 µm x 10 µm for XZ images and 1 µm x 1 µm for the XY images, a scan speed of 30 lines per second and a frame average of 2.

For the image in *Figure 4a* (acquired on a Leica DM6000 microscope with an SP5 scanning unit) we used a pixel size of 1.44 µm x 1 µm (XZ) and a scan speed of 400 lines per second, and a frame average of 12.

## Acknowledgements

This work was supported by the Medical Research Council, grant number MR/K015583/1, to Gail McConnell and John Dempster. Brad Amos is supported by a Leverhulme Emeritus Fellowship. We thank Gillian Robb (SIPBS, University of Strathclyde) for assistance with using Volocity for *Video 2*, Yvonne Vallis (MRC Laboratory of Molecular Biology) for the stained rat brain explant specimens and supply of mouse embryos and Shinya Inoué for the gift of the grating test specimen. We thank Richard Mort, Venkat Venkataraman and Shahida Sheraz (MRC Human Genetics Unit, University of Edinburgh) for providing mouse embryos. Brad Amos and Es Reid declare a competing interest from Mesolens Ltd. The dataset underlying this publication can be accessed via http://pure.strath.ac.uk.

## Additional information

### Competing interests

ER, WBA: Co-founder and shareholder of Mesolens Ltd, a company that specialises in designing and manufacturing optical instruments. The other authors declare that no competing interests exist.

## Funding

| Funder | Grant reference number | Author |
|---|---|---|
| Medical Research Council | MR/K015583/1 | Gail McConnell<br>John Dempster |
| Leverhulme Trust | | William Bradshaw Amos |

The funders had no role in study design, data collection and interpretation, or the decision to submit the work for publication.

## Author contributions

GM, Design and first build of confocal laser scanning system, acquisition of image data, writing the article; JT, Optimising performance of Mesolens in confocal laser scanning mode, preparation of specimens for imaging, acqustion of images, image reconstruction, data analysis, including resolution measurement, writing the article; RA, Acquisition of images and data analysis including resolution measurement, writing the article; JD, Design, testing and implementation of Mesoscan software and electronics to control laser scanning system and data acquisition, writing the article; ER, Designed the objective lens and scan lens, acquired simulated data on lens performance; WBA, Specified the lens design, design and first build of the laser scanning system, data acquisition and analysis, writing the article

## Author ORCIDs

Gail McConnell, http://orcid.org/0000-0002-7213-0686

## Ethics

Animal experimentation: All experimental procedures using animals were conducted in strict accordance with the United Kingdom Animals (Scientific Procedures) Act, 1986 and approved by the Home Office (UK).

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
