## [Decision Letter]

Thank you for submitting your article "A novel optical microscope for imaging large embryos and tissue volumes with sub-cellular resolution throughout" for consideration by *eLife*. Your article has been reviewed by three peer reviewers, and the evaluation has been overseen by a Reviewing Editor and Fiona Watt as the Senior Editor. The following individuals involved in review of your submission have agreed to reveal their identity: Scott Fraser (Reviewer #1); Philipp Keller (Reviewer #2).

The reviewers have discussed the reviews with one another and the Reviewing Editor has drafted this decision to help you prepare a revised submission.

Summary:

McConnell et al. describe the development and characterization of their new Mesolens (and the associated laser scanning confocal microscope) as well as the application of this optical system to the imaging of entire fixed embryonic rat brain slices and E10-E12.5 mouse embryos with relatively high resolution. The Mesolens has unique strengths as it combines the ability to image a large field of view (at low magnification) with an optical design that offers an unusually high numerical aperture: the authors show that their customized confocal microscope equipped with the Mesolens is capable of imaging a field of view up to 6 mm with an axial resolution of about 7 microns. There is currently no commercial system that would allow confocal imaging with comparable optical performance.

The authors' achievements are impressive. The optical design and optical engineering of the Mesolens are cutting-edge and the authors put substantial effort into the characterization and evaluation of their optical system. The manuscript is well written and the presentation of the results is clear and convincing. I believe the development of optical systems with the ability to image large specimens and high resolution (and without the need for tiling and post-hoc image stitching) is not only an important but also a very timely endeavor. Moreover, optical components with such capabilities may be useful not only in the context of confocal microscopy but could also synergize well with other state-of-the-art large-volume imaging techniques, such as light-sheet microscopy.

Essential revisions:

1) There are a couple of other imaging modalities that the authors should mention that can address many of the problems they rightfully raise. Fourier Ptychographic Microscopy solves many of the same problems that the mesolens addresses (see the following paper: Guoan Zheng, Roarke Horstmeyer and Changhuei Yang, "Wide-field, high-resolution Fourier ptychographic microscopy;" Nature Photonics, doi:10.1038/nphoton.2013.187). How does their technique compare to fluorescence Talbot microscopy (see the following paper: Shuo Pang, Chao Han, Mihoko Kato, Paul W. Sternberg, and Changhuei Yang; "Wide and scalable field-of-view Talbot-grid-based fluorescence microscopy;" Optics Letters 37 (23): 5018-5020 (2012))? There are other approaches that are addressing high resolution imaging of the brain with a wide field of view such as the high-throughput serial two-photon tomography published in Nature (Oh, S.W., Harris, J.A., et al. 2014. A mesoscale connectome of the mouse brain. Nature 508, 207-214).

2).The author's dismissal of computer tiling for generating large images at higher resolutions seems unwarranted. Corrections for the inhomogeneity of illumination are much better with tiling (and this is more an issue for bright field rather than fluorescent images) as well as those that can correct for different focus heights across a large field. Also their mesolens is not any faster than tiling, as they point out. They need to better explain the advantages of their mesolens over tiling.

3) The optical throughput in confocal mode was 11%. How does this compare to other lenses? This seems low to me.

4) The mesolens doesn't provide any improvement from other objectives in terms of scattering so it still requires clearing. They say that no existing microsope has all the features of their whole embryos imaged with the mesolens but don't they think that a light sheet microscope could get pretty close with greater speed? Especially with the lower z resolution with would allow for a thicker light sheet with a bigger FOV.

5) How do they protect the lens from corrosive immersion media like BABB?

6) In Figure 4 were the same number of optical sections taken? How does software deconvolution improve things? I'm really impressed by how even the illumination is even deeper in the specimen. Is this just from the clearing or was their an increase in the laser power or exposure/gain as they sectioned deeper? The only exception to this was the acridine orange staining which didn't seem to penetrate into the center of the embryo.

Reviewer #1:

This manuscript outlines the motivation, development, construction and testing of a lens capable of capturing wide field of view and high resolution images, which the authors name the mesolens. This lens is a very much needed adjunct to new whole-mount imaging tools, fostered by the popularization of tissue clearing (BABB, Scale, Clarity) and more recently by expansion microscopy techniques. Independent of a reader's bias for or against any one of these techniques, the need for a lens with high performance and large field of view cannot be argued against.

The authors have generated a manuscript that gives the details and features of the mesolens clearly and cleanly. It is far more than the "advertisement" that other authors might be tempted to create.

The performance is excellent, although the complications and precision in implementing the mesons may limit its broad adoption.

I have only one issue: The throughput of the mesolens is reported as a surprisingly small number, without comment. The authors met say more about how this was measured and put it in context with other lenses. My first impression on seeing the number was that it must be a mistake in execution or in writing.

Reviewer #2:

McConnell et al. describe the development and characterization of their new Mesolens (and the associated laser scanning confocal microscope) as well as the application of this optical system to the imaging of entire fixed embryonic rat brain slices and E10-E12.5 mouse embryos with relatively high resolution. The Mesolens has unique strengths as it combines the ability to image a large field of view (at low magnification) with an optical design that offers an unusually high numerical aperture: the authors show that their customized confocal microscope equipped with the Mesolens is capable of imaging a field of view up to 6 mm with an axial resolution of about 7 microns. There is currently no commercial system that would allow confocal imaging with comparable optical performance.

The authors' achievements are impressive. The optical design and optical engineering of the Mesolens are cutting-edge and the authors put substantial effort into the characterization and evaluation of their optical system. The manuscript is well written and the presentation of the results is clear and convincing. I believe the development of optical systems with the ability to image large specimens and high resolution (and without the need for tiling and post-hoc image stitching) is not only an important but also a very timely endeavor. Moreover, optical components with such capabilities may be useful not only in the context of confocal microscopy but could also synergize well with other state-of-the-art large-volume imaging techniques, such as light-sheet microscopy.

I strongly support the publication of this study. I do not see a need for new experiments or the inclusion of additional data. However, I do have a few minor comments (copied below) that relate to text sections in which clarity could be improved.

Reviewer #3:

The paper from McConnell and workers provides a new low power objective that matches the resolution of higher power objectives, providing improvements in both lateral and axial resolutions. Their mesolens is an impressive piece of technology that is not trivial to manufacturer. They deal very well with the issue of refractive index mismatch and that their mesolens allows for the use of multiple immersion media is both impressive and a huge benefit to researchers. The chromatic corrections of their mesolens are very good. The lower actual versus theoretical z-resolution in confocal mode was surprising. I tend to agree with the authors that their scan lens is the likely culprit. They do a good job in the Introduction of covering other types of lenses used to give a large field of view with high NA.

There are a couple of other imaging modalities that I would like to see them mention that can address many of the problems they rightfully raise. Fourier Ptychographic Microscopy solves many of the same problems that the mesolens addresses (see the following paper: Guoan Zheng, Roarke Horstmeyer and Changhuei Yang, "Wide-field, high-resolution Fourier ptychographic microscopy;" Nature Photonics, doi:10.1038/nphoton.2013.187). How does their technique compare to fluorescence Talbot microscopy (see the following paper: Shuo Pang, Chao Han, Mihoko Kato, Paul W. Sternberg, and Changhuei Yang; "Wide and scalable field-of-view Talbot-grid-based fluorescence microscopy;" Optics Letters 37 (23): 5018-5020 (2012))? There are other approaches that are addressing high resolution imaging of the brain with a wide field of view such as the high-throughput serial two-photon tomography published in Nature (Oh, S.W., Harris, J.A., et al. 2014. A mesoscale connectome of the mouse brain. Nature 508, 207-214).

Here are some technical points I would like to see the authors address:

1) I do take issue with their dismissal of computer tiling for generating large images at higher resolutions in that the corrections for the inhomogeneity of illumination are much better (and this is more an issue for bright field rather than fluorescent images) as well as those that can correct for different focus heights across a large field. Also their mesolens is not any faster than tiling, as they point out. They need to better explain the advantages of their mesolens over tiling.

2) The optical throughput in confocal mode was 11%. How does this compare to other lenses? This seems low to me.

3) The mesolens doesn't provide any improvement from other objectives in terms of scattering so it still requires clearing. They say that no existing microsope has all the features of their whole embryos imaged with the mesolens but don't they think that a light sheet microscope could get pretty close with greater speed? Especially with the lower z resolution with would allow for a thicker light sheet with a bigger FOV.

4) How do they protect the lens from corrosive immersion media like BABB?

5) In Figure 4 were the same number of optical sections taken? How does software deconvolution improve things? I'm really impressed by how even the illumination is even deeper in the specimen. Is this just from the clearing or was their an increase in the laser power or exposure/gain as they sectioned deeper? The only exception to this was the acridine orange staining which didn't seem to penetrate into the center of the embryo.

---

## [Author Response]

*Essential revisions:*

*1) There are a couple of other imaging modalities that the authors should mention that can address many of the problems they rightfully raise. Fourier Ptychographic Microscopy solves many of the same problems that the mesolens addresses (see the following paper: Guoan Zheng, Roarke Horstmeyer and Changhuei Yang, "Wide-field, high-resolution Fourier ptychographic microscopy;" Nature Photonics, doi:10.1038/nphoton.2013.187). How does their technique compare to fluorescence Talbot microscopy (see the following paper: Shuo Pang, Chao Han, Mihoko Kato, Paul W. Sternberg, and Changhuei Yang; "Wide and scalable field-of-view Talbot-grid-based fluorescence microscopy;" Optics Letters 37 (23): 5018-5020 (2012))? There are other approaches that are addressing high resolution imaging of the brain with a wide field of view such as the high-throughput serial two-photon tomography published in Nature (Oh, S.W., Harris, J.A., et al. 2014. A mesoscale connectome of the mouse brain. Nature 508, 207-214).*

Fourier Ptychographic Microscopy (FPM) gives a very wide field of view image with sub-micron resolution, and the Zheng et al. paper published in Nature Photonics is a beautiful demonstration of this 2D imaging method with a very wide field of view. However, the depth of focus of their instrument is purposefully large (~0.3 mm) in order to provide a large tolerance to microscope slide placement errors. This very large depth of focus is not suited to confocal laser scanning microscopy, where a small depth of focus is needed for optical sectioning of the imaged specimen. As such, our instrument was designed to provide a depth of focus around two orders of magnitude less than that of the system by Zheng et al. Also, the low N.A. lens used in this demonstration of wide-field FPM is for use in air only. While this is acceptable for thin tissue sections, restriction to air immersion would give dramatic fall-off in axial resolution performance at depth when imaging thick tissues because of spherical aberration, for which there are presently no correction collars or any other means of compensation. Furthermore, as noted by Zheng et al. in their published work, they state that “the current FPM method is not a fluorescence technique, as fluorescent emission profiles would remain unchanged under angle-varied illumination”. Therefore, while FPM gives high-resolution colour images of thin specimens such as pathology slides, it is not suitable for fluorescence imaging of either thin or thick specimens in either 2D or 3D.

The Reviewing Editor asks on behalf of the Reviewers how our method compares to fluorescence Talbot microscopy, and specifically mentions the paper by Pang et al. As reported in the work of Pang et al., Talbot microscopy usually involves placing the specimen in direct contact with the imaging sensor. As the authors gently explain, this is “not a standard laboratory preparation procedure”, and because of the requirement for immersion and mounting of specimens in high-index material, would be impractical for thick tissue imaging. The work of Pang et al. overcomes this limitation, and makes possible imaging of specimens spatially separated from the sensor with fluorescence contrast and an exceptionally wide-field of view. However, while the lateral performance is quite good (an xy resolution of 1.2 microns, compared with our sub-micron lateral resolution), the reported depth of focus limit of 60: m is too large to be useful for optical sectioning of thick tissue volumes. The imaging speed of 23 seconds of the instrument described by Pang et al. is attractive but the reported unevenness of Talbot spots over an area of 1 mm by 1 mm of 4.73% is very high when compared with both standard wide-field imaging and our wide-field imaging with the Mesolens.

The work of Oh et al. on serial two-photon tomography provides very high lateral resolution images of the fluorescent mouse brain. However, these images require the tissue to be destroyed because of the microtome sectioning needed and this makes repeat imaging impossible. More importantly, stitching and tiling of multiple datasets is required because of the large diameter of the mouse brain.

Figure 2 of Oh, S.W., et al. 2014. A mesoscale connectome of the mouse brain. Nature 508, 207-214, (reproduced in part from this paper to the left, labelled ‘VISp’), displays a coronal section with a centre of injection site is (top), and an example of EGFP-labelled axons in a representative subcortical regions (bottom). However, the stitching and tiling of data has led to a patchwork quilt-like pattern. This is particularly acute in the bottom image, where the datasets are laterally off-set from row to row, and there are dark lines between the datasets where information is not so clearly visible. As shown in the inset to the bottom image, there are also significant differences in signal intensity from one dataset to another. Because of the continuous scanning laser spot that we use with the Mesolens, we do not observe differences in fluorescence signal from one region of the image to another unless it is present in the specimen.

We have amended the Introduction to explicitly mention these other methods:

“Fourier Ptychographic Microscopy (FPM) can give a very wide field of view image with submicro resolution (ref Zheng et al.). […] More importantly, stitching and tiling of multiple datasets is required because of the large diameter of the tissue, and this can result in poor image registration and differences in fluorescence signal from one dataset to another.”

We finally note that some commercial microscopes are described as capable of imaging the same specimen from the macro-scale to the micro-scale (e.g. AZ100 Multizoom, Nikon) but at the low magnifications that permit imaging of large specimens, the N.A. of these microscopes is low, and thus the performance is as for a low N.A. objective. In this paper we present a cure for this problem. We describe a multi-immersion objective lens for wide-field epifluorescence and laser scanning confocal microscopy with a working distance of over 3 mm and a 6 mm FOV that is corrected for a wide range of wavelengths and demonstrate its advantages with large biological specimens.

*2).The author's dismissal of computer tiling for generating large images at higher resolutions seems unwarranted. Corrections for the inhomogeneity of illumination are much better with tiling (and this is more an issue for bright field rather than fluorescent images) as well as those that can correct for different focus heights across a large field. Also their mesolens is not any faster than tiling, as they point out. They need to better explain the advantages of their mesolens over tiling.*

The dataset reproduced above from the paper by Oh et al. suggests that tiling of fluorescence datasets can result in significant inhomogeneity of illumination. While correction is possible, artefacts still remain, in particular for larger inhomogeneities. Some of the recent best performance of stitching and tiling of datasets is reported as “seamless” (Legesse et al., Seamless stitching of tile scan microscope images, J. Microscopy 2015), but it is not without limitation. In this work, Legesse et al. used CARS images obtained from a large biopsy of head and neck squamous cell carcinoma.

However, as evidenced by the original tiled CARS image below of their specimen (A), even after basic correction, image boundaries are still visible. Their method seems to give very good image merging and registration on a large scale (e.g. across the ~6mm of the image) as shown in (C), but no zoom of a small region is presented so it is unclear how accurately fine detail can be registered. We also note that the overall signal level in (C) is decreased by around 25% from that obtained in (A) when applying their method, which may prove limiting when using weakly fluorescent specimens.

Aside from their own algorithm, the work of Legesse et al. also presents a side-by-side comparison of commonly available algorithms for stitching and tiling of the same dataset. Image (D) presents the image after using the ImageJ Grid/Collection stitching plugin, (E) shows the results from using the MosaicJ plugin, and (F) shows the outcome from applying Microscoft ICE. In all instances, image registration is poor and inhomogeneities are clearly visible.

Author response image 1.**DOI:**
http://dx.doi.org/10.7554/eLife.18659.017

We conclude that image merging and registration of tiled datasets may offer some advantage over uncorrected data, but that continuous scanning of a single spot over the same field eliminates this uncertainty altogether. We have amended the Introduction:

“During experiments with laser scanning confocal microscopes in the mid1980s, it became obvious that the optical sectioning, which is the main advantage of the confocal method, did not work with the available lowmagnification objectives, because of their low numerical aperture (N.A.) (1). In specimens such as mouse embryos at the 1012.5 day stage, when the major organs are developing (2), it was impossible to see individual cells in the interior despite the lateral (XY) resolution being sufficient. Since then, stitching and tiling of large datasets has proved to be possible using computercontrolled specimen stages, but this results in a checker board pattern in the final image due to inhomogeneity of illumination and focus height errors which often cannot be corrected by software. Commercial and open source software algorithms are available to perform stitching and tiling but inhomogeneities and problems with dataset registration are clearly visible (ref Legesse et al). Because of the continuous scanning laser spot that we use with the Mesolens, we do not observe differences in fluorescence signal from one region of the image to another unless it is present in the specimen.”

With the Mesolens we also have the possibility of switching from slow scanning mode to fast camera mode with the large field of view in the same focal plane. This is not possible with higher magnification objective lenses. This is of value both in searching for rare image features, and in adjusting the compensation collars of the objective lens. We have amended the Discussion:

“With the Mesolens we also have the possibility of switching from a relatively slow laser scanning mode to fast camera mode with the large field of view in the same focal plane. This is not possible with higher magnification objective lenses. This is of value both in searching for rare image features, and in adjusting the compensation collars of the objective lens.”

*3) The optical throughput in confocal mode was 11%. How does this compare to other lenses? This seems low to me.*

Our reported optical throughput of 11% is the optical throughput of the Mesolens, scan lens and scanning system. To measure the optical throughput we compared the laser power just after the beam splitter in the periscope (see Figure 1 in the manuscript) with the laser power at the specimen plane. A significant loss results from overfilling the top mirror in the periscope, which is required in order to achieve the flattest possible wavefront. There are then additional losses from the scanning mirrors and in the scan lens.

Although the optical throughput of a standard microscope objective can be several tens of percent, the transmission of the complete system is more similar to what we measure here. We also note that the transmission value of the complete system is rarely given explicitly by microscope manufacturers but, based on the quoted power of the laser system (e.g. 50 mW for a KryptonArgon laser operating at a wavelength of 488 nm) and measuring the maximum average power at the specimen plane (e.g. 3 mW after the objective lens), we measure a similar magnitude of optical loss with commercial confocal microscopes compared to the Mesolens. The good transmission of the Mesolens itself is also confirmed by our comparison of the transmission of the Mesolens with that of a 4x/0.1 N.A. lens in camera mode. We also note that the reported value of 11% is the transmission for the laser beam in, and the return image is not clipped on the periscope mirror. We have amended the section

Measurements in optical performance:

“In order to make measurements of the relative optical throughput efficiency of the Mesolens as compared with a commercial lowmagnification, low numerical aperture lens (4x/0.1 N.A. Plan, Nikon) a lightemitting diode (HLMP2855,

Broadcom) was used as a specimen of standard luminosity (10). […] The good transmission of the Mesolens itself is also confirmed by our comparison of the transmission of the Mesolens with that of a 4x/0.1 N.A. lens in camera mode.”

*4) The mesolens doesn't provide any improvement from other objectives in terms of scattering so it still requires clearing. They say that no existing microsope has all the features of their whole embryos imaged with the mesolens but don't they think that a light sheet microscope could get pretty close with greater speed? Especially with the lower z resolution with would allow for a thicker light sheet with a bigger FOV.*

Work by Jaehrling et al., 3DVisualization of nerve fiber bundles by ultramicroscopy, Medical Laser Application, 2008, presents lightsheet imaging of fluorescent labelled E12.5 mouse embryos that were cleared using the same BABB method as used in our work. We note that the authors used a lens with 4x magnification and an unusually high numerical aperture (N.A.=0.28), which is no longer commercially available in the UK and we have been unable to obtain performance data for this lens. As part of their study the authors modelled the spatial intensity profiles of lightsheets for various fields of view.

However, as stated by the authors, it is impossible to produce a lightsheet wide enough for such specimens with a thickness of less than 50 microns. This is an order of magnitude less than the axial resolution of the Mesolens. A lightsheet with a thickness comparable to the axial resolution of the Mesolens would only cover a field much smaller than 6 mm. As the Reviewing Editor and Reviewers correctly suggest though, by significantly compromising the axial resolution to a value of many tens of microns, lightsheet imaging would be possible.

Nevertheless, we expect that the Mesolens used as the collecting objective lens for lightsheet microscopy would be a useful way of extending lightsheet microscopy, rather than merely an equivalent technology. For example, the high collection efficiency (about 4 times greater than the x4/0.28 N.A. lens used by Jaehrling et al.) would be of great benefit for studying weakly labeled specimens while providing the advantage of reduced photobleaching and phototoxicity offered by standard lightsheet microscopy.

*5) How do they protect the lens from corrosive immersion media like BABB?*

Because embryos may be examined in a variety of optically dissimilar fluids such as water, glycerol, oil and benzyl benzoate (such as for the clearing liquid BABB), it was necessary to make the lens suitable for use with different immersion fluids and mounting media. The Mesolens was designed for immersion into noncorrosive immersion media such as oil (Type DF), water and glycerol. We use BABB only as a mounting medium, and the specimen and BABB are separated from the immersion fluid by a type 1.5 coverslip.

*6) In Figure 4 were the same number of optical sections taken? How does software deconvolution improve things? I'm really impressed by how even the illumination is even deeper in the specimen. Is this just from the clearing or was their an increase in the laser power or exposure/gain as they sectioned deeper? The only exception to this was the acridine orange staining which didn't seem to penetrate into the center of the embryo.*

We have amended the manuscript in the Imaging of biological specimens and Discussion:

“To demonstrate the superior sectioning capability of the Mesolens compared to a standard objective lens with comparable magnification, we imaged the same cleared and acridine orange stained 10 day old mouse embryo with the Mesolens and with a dry objective lens of similar magnification (5x/0.15 N.A. HCX PL Fluotar, Leica Microsystems) mounted on an upright laser scanning microscope. The embryo was lying on its side so that XY scanning produced an optical section that was anatomically sagittal. The XZ images are displayed in Figure 4. […] We did not adjust the laser power or detector gain when imaging deeper into tissue.”

*Reviewer #1:*

*This manuscript outlines the motivation, development, construction and testing of a lens capable of capturing wide field of view and high resolution images, which the authors name the mesolens. This lens is a very much needed adjunct to new whole-mount imaging tools, fostered by the popularization of tissue clearing (BABB, Scale, Clarity) and more recently by expansion microscopy techniques. Independent of a reader's bias for or against any one of these techniques, the need for a lens with high performance and large field of view cannot be argued against.*

*The authors have generated a manuscript that gives the details and features of the mesolens clearly and cleanly. It is far more than the "advertisement" that other authors might be tempted to create.*

*The performance is excellent, although the complications and precision in implementing the mesons may limit its broad adoption.*

*I have only one issue: The throughput of the mesolens is reported as a surprisingly small number, without comment. The authors met say more about how this was measured and put it in context with other lenses. My first impression on seeing the number was that it must be a mistake in execution or in writing.*

We have responded to this comment above in our response to the summary provided by the Reviewing Editor.

*Reviewer #3:*

*There are a couple of other imaging modalities that I would like to see them mention that can address many of the problems they rightfully raise. Fourier Ptychographic Microscopy solves many of the same problems that the mesolens addresses (see the following paper: Guoan Zheng, Roarke Horstmeyer and Changhuei Yang, "Wide-field, high-resolution Fourier ptychographic microscopy;" Nature Photonics, doi:10.1038/nphoton.2013.187). How does their technique compare to fluorescence Talbot microscopy (see the following paper: Shuo Pang, Chao Han, Mihoko Kato, Paul W. Sternberg, and Changhuei Yang; "Wide and scalable field-of-view Talbot-grid-based fluorescence microscopy;" Optics Letters 37 (23): 5018-5020 (2012))? There are other approaches that are addressing high resolution imaging of the brain with a wide field of view such as the high-throughput serial two-photon tomography published in Nature (Oh, S.W., Harris, J.A., et al. 2014. A mesoscale connectome of the mouse brain. Nature 508, 207-214).*

We have responded to this comment above in our response to the summary provided by the Reviewing Editor.

*Here are some technical points I would like to see the authors address:*

*1) I do take issue with their dismissal of computer tiling for generating large images at higher resolutions in that the corrections for the inhomogeneity of illumination are much better (and this is more an issue for bright field rather than fluorescent images) as well as those that can correct for different focus heights across a large field. Also their mesolens is not any faster than tiling, as they point out. They need to better explain the advantages of their mesolens over tiling.*

*2) The optical throughput in confocal mode was 11%. How does this compare to other lenses? This seems low to me.*

We have responded to this comment above in our response to the summary provided by the Reviewing Editor.

*3) The mesolens doesn't provide any improvement from other objectives in terms of scattering so it still requires clearing. They say that no existing microsope has all the features of their whole embryos imaged with the mesolens but don't they think that a light sheet microscope could get pretty close with greater speed? Especially with the lower z resolution with would allow for a thicker light sheet with a bigger FOV.*

We have responded to this comment above in our response to the summary provided by the Reviewing Editor.

*4) How do they protect the lens from corrosive immersion media like BABB?*

We have responded to this comment above in our response to the summary provided by the Reviewing Editor.

*5) In Figure 4 were the same number of optical sections taken? How does software deconvolution improve things? I'm really impressed by how even the illumination is even deeper in the specimen. Is this just from the clearing or was their an increase in the laser power or exposure/gain as they sectioned deeper? The only exception to this was the acridine orange staining which didn't seem to penetrate into the center of the embryo.*

We have responded to this comment above in our response to the summary provided by the Reviewing Editor.